# Evaluating the diagnostic utility of the slide agglutination method for brucellosis in Saudi Arabia: A retrospective study at International Medical Center, Jeddah, Saudi Arabia

Reham Kaki[1,2,3,4]*, Hala Zeinelabidin[3], Sukainah N. Rashed[5], Asmaa Baba[6]

1 Department of Medicine, King Abdulaziz University, Jeddah, Saudi Arabia, 2 Department of Infectious Disease, King Abdulaziz University, Jeddah, Saudi Arabia, 3 Department of Medicine, International Medical Center, Jeddah, Saudi Arabia, 4 Department of Infectious Disease, International Medical Center, Jeddah, Saudi Arabia, 5 Department of Pathology and Laboratory Medicine and Blood Bank, International Medical Center, Jeddah, Saudi Arabia, 6 Department of microbiology, International Medical Center, Jeddah, Saudi Arabia

* rmkaki@kau.edu.sa

## Abstract

### Background

Brucellosis, a prevalent zoonotic disease in endemic regions like Saudi Arabia, poses diagnostic challenges due to its nonspecific clinical manifestations. This disease is endemic in various regions, including the Mediterranean basin, the Middle East, Asia, Africa, and Latin America. This retrospective study is designed to evaluate the diagnostic utility of the slide agglutination method for brucellosis. By examining its sensitivity, specificity, and overall performance, this research will provide insights into the reliability of the slide agglutination method as a diagnostic tool in a high-prevalence setting.

### Methods

This retrospective study, conducted at the International Medical Center, Jeddah, compared the real-world diagnostic performance of the Slide Agglutination Method (SAM) with gold-standard blood cultures and Brucella serology. Over 734 cases (37 Brucella-positive and 697 Brucella-negative) were reviewed between 2019 and 2022.

### Results

SAM achieved 86.5% sensitivity and 99.3% specificity, with an average turnaround time of 2.32 hours. In contrast, blood culture showed 40.9% sensitivity and 94.5% accuracy, and conventional serology demonstrated comparable sensitivity but longer processing times.

**Data availability statement:** All relevant data are within the paper and its Supporting Information files.

**Funding:** The author(s) received no specific funding for this work.

**Competing interests:** The authors have declared that no competing interests exist.

## Conclusion

SAM combines speed, simplicity, and robust accuracy, making it an ideal first-line screen for acute brucellosis in endemic, resource-limited settings. However, its performance in chronic or low-titer infections remains less well characterized. Our detailed operational metrics fill a local evidence gap and support the Saudi Ministry of Health in updating national screening guidelines.

---

### Author summary

Brucellosis is a disease that spreads from animals to humans and remains a significant health problem in many parts of the world, including Saudi Arabia. Diagnosing brucellosis can be difficult because its symptoms, such as fever, are similar to many other diseases. In this study, we evaluated the effectiveness of a simple and quick diagnostic test called the slide agglutination method (SAM). We compared the slide agglutination method to more advanced methods like blood cultures, which is also considered the gold standard, and specialized serology tests to understand how well it works in real-world conditions.

Our findings show that the slide agglutination method is highly sensitive in detecting brucellosis and provides results in just over two hours. This makes it much faster than other traditional methods. Even though it works well for initial screening, confirmatory tests are still needed, especially in chronic cases. By integrating SAM with other diagnostic tools, healthcare providers can diagnose brucellosis more effectively and improve outcomes for patients.

## Introduction

Brucellosis is a zoonotic infection caused by the Gram-negative bacterium Brucella, transmitted primarily through direct contact with infected animals or consumption of contaminated animal products [1,2]. This disease is endemic in various regions, including the Mediterranean basin, the Middle East, Asia, Africa, and Latin America, presenting a significant public health challenge due to its varied and nonspecific clinical manifestations such as fever, sweats, malaise, and muscle pain [1,3]. In Saudi Arabia, the infection rate reaches approximately 70 cases per 100,000 people, with morbidity varying by region [4].

The diagnosis of brucellosis relies heavily on laboratory testing due to the nonspecific nature of its symptoms. The gold standard for diagnosis is bacterial culture, known for its high specificity but limited by its time-consuming nature and variable sensitivity. Consequently, serological tests like rapid plate agglutination tests and ELISA are commonly used due to their rapid results and ease of use [5,6]. The Slide Agglutination Method, using Omega reagents, is one such serological test employed for its rapid, qualitative detection of Brucella-specific antibodies. This method has

shown sensitivity ranges comparable to other agglutination-based tests (85–100%), but like other serological methods, its specificity may be reduced in individuals with repeated exposure to Brucella or those with a history of the disease [5]. Molecular diagnostic methods, such as polymerase chain reaction (PCR), have emerged as highly sensitive and specific alternatives for detecting Brucella DNA. These methods are particularly useful in chronic cases where serological tests might not be as reliable. Despite their advantages, the high cost and limited availability of molecular tests pose challenges for their widespread use in resource-limited settings [7].

This retrospective study is designed to evaluate the diagnostic utility of the Slide Agglutination Method (SAM) for brucellosis at the International Medical Center in Jeddah, Saudi Arabia. We hypothesize that SAM will demonstrate high sensitivity and specificity compared with the reference standard. By examining these parameters and overall performance, this research will provide insights into the reliability of SAM as a rapid diagnostic tool in clinical practice.

## Materials and methods

### Ethics statement

The authors confirm that the ethical policies of the journal, as noted on the journal's author guidelines page, have been adhered to. Ethical approval was obtained from the Institutional Review Board (IRB) at the International Medical Center, Jeddah, Saudi Arabia, with the IRB approval number IMC-IRB #2023-04-208, and NCBE Registration No: (H-02-J-010).

### Study design and population

This retrospective observational study was conducted at the International Medical Center (IMC) in Jeddah, Saudi Arabia, to assess the real-world diagnostic performance of the Slide Agglutination Method using Omega reagents compared to blood culture and Brucella serology in detecting brucellosis. The study included all patients aged 18 years or older who presented with symptoms indicative of brucellosis and underwent Slide Agglutination Method testing, blood culture, or serology between January 2019 and December 2022.

The study population comprised both acute and chronic brucellosis cases. Chronic brucellosis was defined as symptom duration of one year or longer [8]. These chronic cases were included to mirror routine clinical practice, recognizing that serological titers and culture yields can be lower in chronic infection [9,10]. By retaining chronic presentations, we ensure our evaluation shows the full spectrum of diagnostic challenges encountered in a retrospective serology-based study population.

Information extracted from electronic health records included demographic data, clinical presentation, laboratory results, and treatment details.

### Sample size

The required sample size for this study was determined using a power analysis conducted with G*Power version 3.1.9.4. The analysis was designed to ensure adequate statistical power to obtain a significant correlation effect of 0.95. Using an alpha coefficient of 0.05, the study deemed it necessary to have at least 595 participants. This sample size was sufficient in a way that it would minimize chances of Type II errors.

### Microbiological testing

**Blood culture.** Venous blood (8–10 mL per bottle, collected aseptically before antibiotic administration) was inoculated into both aerobic and anaerobic bottles of an automated culture system (e.g., BacT/ALERT or VersaTREK). Initially, all bottles were incubated at 35 °C and monitored continuously for 7 days. Bottles with no positive signal at day 7 were subjected to extended incubation for up to 10 days to enhance the recovery of slow-growing organisms. At the end of the 14 days, all bottles that remained negative were subcultured (terminal/blind) onto blood, MacConkey, chocolate, and

Brucella-selective agar. Bottles that signaled positive at any point were immediately subcultured and examined by Gram stain, colony morphology on Brucella agar, and oxidase/urease testing. This extended culture method improves the diagnostic sensitivity for brucellosis, despite the increased turnaround time.

**Serology.** Brucella IgG and IgM antibodies were detected using ELISA (Vircell Reagents). ELISA provided quantitative measurements and differentiation between acute (IgM) and chronic (IgG) infections, with results available within 24–48 hours. In line with previous literature [11], antibody titers ≥25 IU/mL for IgG or ≥20 IU/mL for IgM were considered positive. A positive result for either IgM or IgG alone was sufficient to indicate Brucella infection. In previous literature, IgG or IgM positivity yielded a sensitivity of 94.1% and a specificity of 97.1% [12].

**Slide Agglutination Method (SAM).** The rapid qualitative test was conducted using the Micropath Rose Bengal reagent (Omega Diagnostics, Ref OD265), following manufacturer instructions. 50 µL of serum was mixed with the antigen on a glass slide and rotated for 4 minutes. Agglutination was visually assessed immediately after mixing. A positive reaction was defined as visible agglutination corresponding to an antibody concentration of ≥25 IU/mL, typically consistent with a titer of ≥1:80. This interpretation threshold aligns with previous literature, which considers agglutination at 1:80 or higher as positive for Brucella infection [13,14]. Serial twofold dilutions of serum samples were prepared to determine endpoint titers. Each test followed SAM's standardized operating protocol to ensure consistency and repeatability. While effective for initial screening, limitations included potential false negatives in chronic cases and false positives due to cross-reactivity with other pathogens.

## Statistical analysis

Data were checked for correctness, completeness, and normality using the Shapiro-Wilk test. Categorical variables were presented as frequencies and percentages, while numerical variables were presented as mean ± standard deviation. An independent samples t-test was used to compare the turnaround times of the Slide Agglutination Method, blood culture, and antibody testing between Brucella-positive and -negative cases. Statistical analyses were performed using SPSS version 24.0, with a 95% confidence interval and a significance level of $p < 0.05$. The diagnostic performance of blood culture, serology, and the slide agglutination method was evaluated using standard parameters, including sensitivity, specificity, positive predictive value (PPV), negative predictive value (NPV), and accuracy.

$$Sensitivity\ (\%) = \frac{True\ positives}{True\ positives + False\ negatives}\ x\ 100$$

$$Specificity\ (\%) = \frac{True\ negatives}{True\ negatives + Flase\ positives}\ x\ 100$$

$$PPV\ (\%) = \frac{True\ positives}{True\ Positives + False\ positives}\ x\ 100$$

$$NPV\ (\%) = \frac{True\ negatives}{True\ negatives + False\ negatives}\ x\ 100$$

$$Accuracy\ (\%) = \frac{True\ positives + True\ negatives}{Total\ cases}\ x\ 100$$

## Results

A total of 734 cases were included in this retrospective chart review study. Among them, 697 (95%) were Brucella negative and 37 (5%) were Brucella positive.

The most common clinical feature of Brucella infection was fever (97.3%), followed by malaise (89.2%). The least common feature was abdominal pain, present in 21.6% of cases (Fig 1). The commonest type of Brucella infection was co-infection with *B. abortus* and *B. melitensis*, observed in 64.9% of the Brucella-positive cases (Fig 2). About 40.5% of the Brucella-positive cases were cured. The outcome for 21 (56.8%) cases was unknown due to loss to follow-up or incomplete documentation in the retrospective chart review (Fig 3).

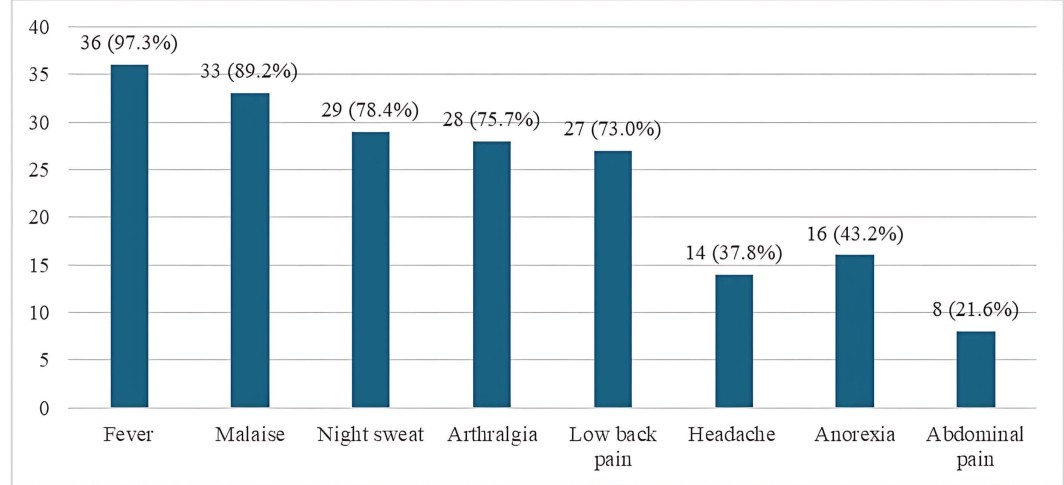

**Fig 1. Clinical features among Brucella-positive cases (n = 37).**

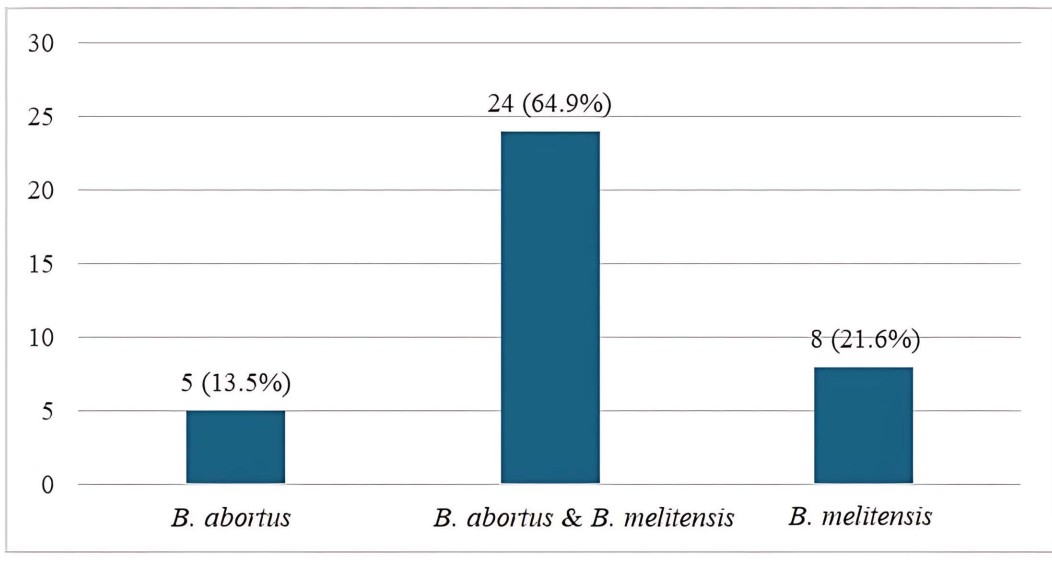

**Fig 2. Type of Brucella in Brucella positive cases (n = 37).**

Table 1 compares the turnaround times between Brucella-positive and Brucella-negative cases across three diagnostic methods. The slide agglutination method showed a significantly faster turnaround for Brucella-positive cases (2.32±1.53 hours) compared to Brucella-negative cases (3.32±2.41 hours), with a p-value of 0.013. Blood culture had a significantly longer turnaround for Brucella-positive cases (151.00±31.44 hours) compared to Brucella-negative cases (119.29±45.81 hours), with a p-value of 0.002. In contrast to the other two tests for diagnosing Brucellosis, the antibody test did not show a statistically significant difference in turnaround time between Brucella-positive (88.81±70.92 hours) and Brucella-negative cases (73.78±58.02 hours), with a p-value of 0.174.

All Brucella-positive cases consumed unpasteurized animal products except one case. Most of them presented with no focal disease, 33 (89.2%). One case had osteoarticular disease (2.7%), and three cases had spondylitis (8.1%). Most of the Brucella-positive cases were treated with doxycycline+rifampicin combination, 29 (78.4%) (Table 2). The mean treatment duration was 6.81±1.53 weeks. The mean symptom duration was 16.68±29.90 days among Brucella-positive cases.

Table 3 compares the diagnostic performance of blood culture, serology, and the slide agglutination method. Serology showed the highest sensitivity (90.3%), followed by the slide agglutination method (86.5%). Both serology and the slide agglutination method had an accuracy of 99.3% and demonstrated perfect specificity (100%) and positive predictive value (PPV). Blood culture also showed perfect specificity and PPV but had the lowest sensitivity (40.9%) and accuracy (94.5%).

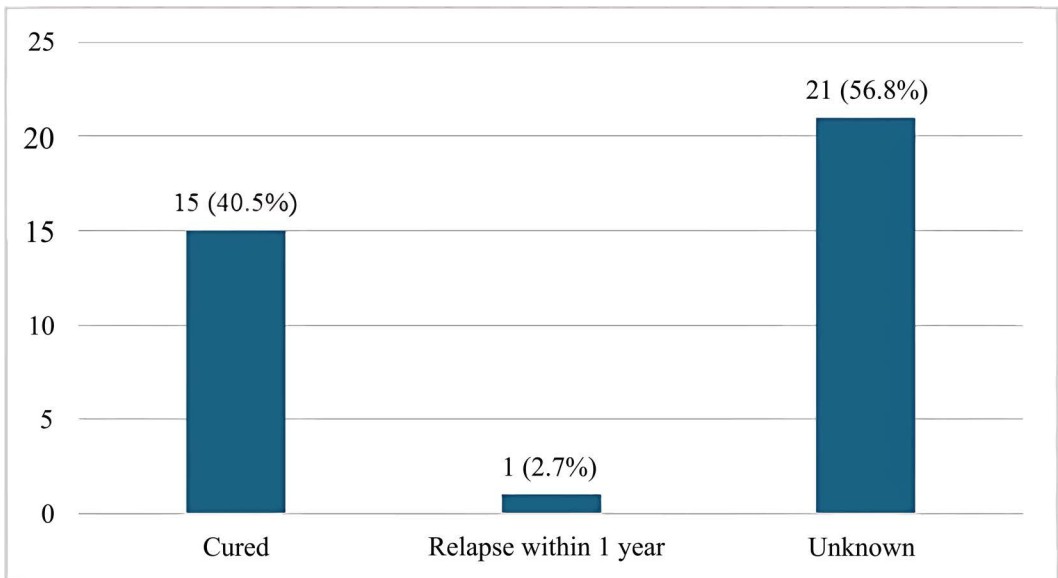

**Fig 3. Outcome of Brucella positive cases (n=37).**

**Table 1. Turnaround time comparison between Brucella positive and negative cases (in hours).**

| Turn around | Brucella Positive (mean±SD) | Brucella Negative (mean±SD) | t-value | p-value | 95% CI |
|---|---|---|---|---|---|
| Slide Agglutination Method | 2.32±1.53 | 3.32±2.41 | 2.497 | .013 | 1.78-0.21 |
| Blood culture | 151.00±31.44 | 119.29±45.81 | -3.160 | .002 | -11.92-51.48 |
| Antibody test | 88.81±70.92 | 73.78±58.02 | -1.361 | .174 | 6.68-36.73 |

## Discussion

This study evaluated the performance of the Slide Agglutination Method (SAM) using Omega reagents for diagnosing brucellosis at the International Medical Center in Jeddah, Saudi Arabia, compared to traditional diagnostic methods, including blood culture and serology. Our results showed that the SAM has an average turnaround time of 2.32 hours for Brucella-positive cases. Such rapid diagnostic tests for infectious diseases not only help prevent the spread of infections but also improve patient outcomes and support antimicrobial stewardship [15]. The clinical features of Brucella infection observed in this study were similar to those in prior reports, which showed fever as one of the most common symptoms [16]. Many Brucella-positive cases showed laboratory findings like anemia, which is consistent with the study by Akdeniz et al. [17], who also reported similar hematologic abnormalities.

In terms of diagnostic performance for brucellosis, the slide agglutination method showed a sensitivity and accuracy of 86.5% and 99.3%, respectively. Other studies have reported sensitivities between 87–98% [18,19,20], likely showing protocol variations. Reda et al. [19] followed the BioMed-Rose Bengal manufacturer's instructions exactly (50 µL serum mixed with 50 µL antigen, 4 minutes' rotation), leading to 96% sensitivity. On the other hand, Díaz et al. [20] used the Veterinary Laboratory Agency (VLA) RBT antigen with an extended 8-minute rocking protocol and reported 87.4% sensitivity. We utilized the Micropath Rose Bengal reagent (Omega Diagnostics, Ref OD265) with a ≥ 1:80 titer cut-off during a 4-minute rotation, yielding 86.5% sensitivity. A recent meta-analysis showed a pooled sensitivity of 89.7% [95% CI: 82.0–94.4] and

**Table 2. Characteristics of Brucella-positive cases by Exposure, Treatment Regimen, and Focal Disease.**

| Variable | Category | N (%) |
|---|---|---|
| **Unpasteurized Animal Products** | Yes | 36 (97.3) |
| | No | 1 (2.7) |
| **Treatment** | Ciprofloxacin + Rifampin | 1 (2.7) |
| | Doxycycline + Ceftriaxone | 2 (5.4) |
| | Doxycycline + Rifampin | 29 (78.4) |
| | Doxycycline + Rifampin + Gentamicin | 1 (2.7) |
| | Doxycycline + Rifampin + Ceftriaxone | 3 (8.1) |
| | Rifampin | 1 (2.7) |
| **Focal Disease** | Osteoarticular disease | 1 (2.7) |
| | Spondylitis | 3 (8.1) |
| | None | 33 (89.2) |

**Table 3. Comparison of parameters among different techniques.**

| Parameters Assessed | Blood Culture | Serology | Slide Agglutination Method |
|---|---|---|---|
| **True positive (n)** | 9 | 28 | 32 |
| **True negative (n)** | 213 | 375 | 696 |
| **False Positive (n)** | 0 | 0 | 0 |
| **False negative (n)** | 13 | 3 | 5 |
| **Sensitivity (%)** | 40.9 | 90.3 | 86.5 |
| **Specificity (%)** | 100 | 100 | 100 |
| **Positive Predictive Value (PPV) (%)** | 100 | 100 | 100 |
| **Negative Predictive Value (NPV) (%)** | 94.2 | 99.2 | 99.3 |
| **Accuracy (%)** | 94.5 | 99.3 | 99.3 |

specificity of 94.1% [95% CI: 83.1–98.1] for Rose Bengal tests, with high heterogeneity ($I^2 = 70.8$ for sensitivity and 90.3 for specificity) which might be due to methodological differences [21]. Such heterogeneity can also be due to the differences in disease stage since SAM's performance tends to change in chronic cases, as shown by Megersa et al. [22] and Koyuncu et al. [23].

When placed alongside reported sensitivities of ELISA (e.g., Xu et al. [24] reported 98.8% sensitivity) and PCR-based assays (Al-Attas et al. [25] reported 100% sensitivity), our findings establish a clear performance benchmark for SAM. This advances our understanding of brucellosis diagnostics by showing that SAM's operational advantages, such as speed, simplicity, and reliability, can streamline laboratory workflows and guide method selection in both endemic and resource-limited settings.

Even though the SAM demonstrated high specificity and positive predictive value (PPV), similar to findings by Sathyanarayan et al. [26], its limitations in terms of sensitivity in certain clinical presentations demonstrate the need for confirmatory tests, especially in patients with chronic brucellosis. Serology also demonstrated a high sensitivity and accuracy, which is again in line with results observed by Sathyanarayan et al. [26].

Although blood culture is the gold standard, it demonstrated a lower sensitivity (40.9%) and accuracy (94.5%) compared to other methods. This might be due to the nature of brucellosis in its early stages, where patients show persistent low-level bacteremia that can be detected through multiple blood culture samples [27]. However, as the disease progresses, the bacteria are cleared from the bloodstream and accumulate in macrophages. This results in a decreased bacterial load in circulation. This decrease, together with the less consistent nature of the bacteremia, can make it more difficult to isolate the pathogen [27]. Therefore, culture is not very sensitive during the later stages of the disease [28,29].

This study was conducted in a real-world setting in an endemic area, providing practical insights into the Slide Agglutination Method's effectiveness and usability as a diagnostic tool. This context enhances the generalizability of findings to other high-burden regions. Additionally, the study's comparative design across diagnostic methods allows for a valuable assessment of the Slide Agglutination Method's advantages in terms of speed and sensitivity, particularly in acute cases.

This study's evaluation of SAM's 2.32-hour turnaround, 86.5% sensitivity, and 99.3% specificity in a Saudi tertiary-care setting fills a local evidence gap in operational diagnostics. By quantifying these performance metrics under real-world laboratory conditions, the data supply the Saudi Ministry of Health (MOH) with insights to update screening guidelines, incorporate rapid reporting through the Health Electronic Surveillance Network (HESN), and prioritize resource allocation for strengthening national brucellosis control efforts.

Despite its many strengths, several limitations must be considered. Even though SAM showed favorable performance in the Brucella-endemic area like Saudi Arabia, its diagnostic utility in non-endemic areas may be more limited due to differences in disease prevalence and clinical presentation. In such areas, clinicians should consider epidemiological factors and use SAM results alongside clinical evaluation and confirmatory testing to avoid misdiagnosis. Moreover, the study's retrospective design may introduce biases related to record accuracy and completeness. Chronic brucellosis cases were not distinguished in the dataset, therefore, the sensitivity of the Slide Agglutination Method (SAM) in chronic presentations could not be analyzed separately. This limits the study's ability to determine SAM's diagnostic performance across the full clinical spectrum of brucellosis. Additionally, as this research was conducted within a single healthcare center, the findings may be less generalizable across different healthcare settings.

## Conclusion

This study confirms that the Slide Agglutination Method is an effective, rapid, and cost-efficient screening tool for brucellosis in Saudi Arabia, where the infection rate reaches 70 cases per 100,000 people, by enabling prompt case detection in resource-limited, high-burden settings. Our detailed operational metrics fill a local evidence gap and support the Saudi Ministry of Health in updating national screening guidelines. However, this study did not distinguish between chronic or low-titer cases and acute cases, so its performance in such contexts remains uncharacterized. These limitations

demonstrate the necessity of incorporating confirmatory assays such as PCR for accurate diagnosis and demonstrate the need for future studies that evaluate combined agglutination-molecular protocols in different disease stages and geographic hotspots to improve context-specific public health interventions.

## Supporting information

**S1 Raw Data. Raw data for the study.**
(XLSX)

**S1 Dataset. SPSS dataset.**
(CSV)

**S1 Checklist. Strobe checklist.**
(DOC)

## Acknowledgments

The author is very thankful to all the associated personnel in any reference that contributed to/this research.

## Author contributions

**Conceptualization:** Hala Zeinelabidin, Sukainah N. Rashed, Asmaa Baba.

**Formal analysis:** Reham Kaki, Asmaa Baba.

**Supervision:** Reham Kaki.

**Validation:** Reham Kaki, Sukainah N. Rashed.

**Writing – original draft:** Reham Kaki.

**Writing – review & editing:** Reham Kaki, Hala Zeinelabidin, Sukainah N. Rashed, Asmaa Baba.

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
