## [Decision Letter · Decision Letter 0]

Response to Reviewers
Revised Manuscript with Track Changes
Manuscript

Shaden Kamhawi

co-Editor-in-Chief

Paul Brindley

co-Editor-in-Chief

**Additional Editor Comments (if provided):**
**Journal Requirements:**
**Reviewers' comments:**

**Key Review Criteria Required for Acceptance?**

**Methods**

-Are the objectives of the study clearly articulated with a clear testable hypothesis stated?

-Is the study design appropriate to address the stated objectives?

-Is the population clearly described and appropriate for the hypothesis being tested?

-Is the sample size sufficient to ensure adequate power to address the hypothesis being tested?

-Were correct statistical analysis used to support conclusions?

-Are there concerns about ethical or regulatory requirements being met?

Reviewer #1: This was a retrospective observational study to evaluate the diagnostic performance of the Slide Agglutination Method (SAM). The investigators wanted to access the sensitivity, specificity, and overall performance of SAM compared to the gold standard blood cultures and Brucella serology. On that score the objective is well stated and clear. The design is very simple, all records of people aged 18 years and older who reported to the health facility from 2019 to 2022 were analyzed.

Sample size was not calculated and mentioned so there is no way to tell. There is evidence of ethical approval.

Based on the objectives, the statistical analysis done is adequate

It is stated that A Standard Wright test (SWT) ≥ 160 or a tube agglutination test with antihuman globulin

(AHG TAT) ≥ 160 was used as the reference standard. Can they clarify this? Was this done for this study with archived blood samples or this was also part for the data retrieved for analysis in this study.

Reviewer #2: Question 1: Not clear blood culture specific operation details, such as acquisition, training time is extended to 14 days in order to improve the sensitivity or higher).

Question 2: SAM's interpretation standards (such as "agglutination 1:8 0 or more positive") are consistent with the reagent instruction? Additional quotes or explanations are required.

Suggestion: replenish blood culture and SAM's standardized operation process, ensure the method repeatability.

Reviewer #3: 1.-Are the objectives of the study clearly articulated with a clear testable hypothesis stated?

The objectives are clear but the hypothesis is not specifically formulated.

2.-Is the study design appropriate to address the stated objectives?

The authors do not specify the antibody titers for serology tests (IgG or IgM) that are considered positive in the Materials and Methods section. They also do not clarify whether a positive result for IgM or IgG alone is considered positive or whether positivity for both is necessary for the diagnosis.

The authors also state that serology offers high sensitivity and specificity, but they do not quantify this sensitivity or specificity or provide any citations to support their claim (line 112). It should be noted that other publications have described false positives in non-endemic regions.

3.-Is the population clearly described and appropriate for the hypothesis being tested?

Regarding the patient sample, were some patients with chronic brucellosis? The authors should clarify and justify this point. The accuracy of serology and cultures may vary in cases of chronic brucellosis. On the other hand, it would be strange if in a retrospective sample based on serology tests there were no cases of chronic brucellosis.

4.-Is the sample size sufficient to ensure adequate power to address the hypothesis being tested?

No sample size calculation is performed.

5.-Were correct statistical analysis used to support conclusions?

The authors should clarify when a IgG or IgM serology was considered positive.

6.-Are there concerns about ethical or regulatory requirements being met?

Yes

**Results**

-Does the analysis presented match the analysis plan?

-Are the results clearly and completely presented?

-Are the figures (Tables, Images) of sufficient quality for clarity?

Reviewer #1: The results are are well presented and the tables are clear

Reviewer #2: Question 3: In table 4 "True positive (%)" column value conflict (such as blood culture of "True positive = 24.3%" and "Sensitivity = 40.9%" logic). Need to recalibrate calculations or amend tables.

Question 4: Did not talk about SAM's performance in chronic cases (only mention "should be combined with other detection" but did not provide data support).

Suggestion: correct form, complement calculation formula in detail. If chronic cases were included in the study, the sensitivity of SAM should be analyzed separately. If not, it should be clearly stated in the limitation.

Reviewer #3: 7.-Does the analysis presented match the analysis plan?

Yes

8.-Are the results clearly and completely presented?

The results section begins by describing the turnaround time of the different techniques. It is probably better to begin with a description of the patient sample, their symptoms, risk factors, and treatment. (line 167 et seq.).

Why is the outcome unknown for 56.8% of cases?

9.-Are the figures (Tables, Images) of sufficient quality for clarity?

Table 3 is irrelevant. What is already stated in the text is sufficient.

Figure 1 contributes nothing to the article's content and can be deleted.

Figure 3 is irrelevant and can be deleted.

**Conclusions**

-Are the conclusions supported by the data presented?

-Are the limitations of analysis clearly described?

-Do the authors discuss how these data can be helpful to advance our understanding of the topic under study?

-Is public health relevance addressed?

Reviewer #1: Based on the objectives, the conclusions are justified . The limitations of the analysis are not clearly articulated however, the study does not provide any new insights other than what is already known or what is expected hence the public health relevance is not clear. The authors could provide some context of the prevalence of brucellosis in the study region to make a case for the public health relevance.

Reviewer #2: Question 5: Not compared with similar studies (such as 10-12 only general references mentioned "sensitivity", but not concrete analysis the reasons for differences).

Question 6: low sensitivity of blood cultures (40.9%), insufficient explanation (such as whether because of sampling time, antibiotic use, or culture differences?) .

Suggestion: Increase compared with recent research (e.g., Meta data analysis or area), heterogeneity of SAM performance are discussed. The reasons for the low sensitivity of blood culture (such as disease stage and operational factors) were analyzed in combination with the literature.

Reviewer #3: 10.-Are the conclusions supported by the data presented?

In the conclusion, the authors state that "…confirmatory testing remains essential for comprehensive diagnosis, especially in chronic cases" (lines 250-251). This statement is not supported by the data provided by the authors in this study.

In the discussion section (lines 207-210), it is stated that "Our results showed that the SAM has an average turnaround time of 2.32 hours for Brucella-positive cases, which is significantly faster than blood culture..." This sentence is inappropriate since the authors do not compare techniques in the results but compare positive and negative results for the same technique.

11.-Are the limitations of analysis clearly described?

The results could be helpful in endemic areas, but nothing is mentioned regarding non-endemic areas

12.-Do the authors discuss how these data can be helpful to advance our understanding of the topic under study?

The authors should discuss this point better and refer to the results of more studies on the outcome of serologies and other laboratory methods.

13.--Is public health relevance addressed?

Yes

**Editorial and Data Presentation Modifications?**

Reviewer #1: Line 171: B. abortus and B. melitensis (should be in italics), same with figure 4

Table 2: The frequency and percentage can be in the same column eg 36 (97.3%)

Reviewer #2: MInor Revision

Reviewer #3: (No Response)

**Summary and General Comments**

Reviewer #1: The paper is very well written and the objectives are clearly stated. The novelty and significance has not been well articulated. What new information is this study adding to the already existing knowledge on the subject matter and how is this going to change policy in Saudi Arabia?

Reviewer #2: No

Reviewer #3: (No Response)

PLOS authors have the option to publish the peer review history of their article (what does this mean? ). If published, this will include your full peer review and any attached files.

**Do you want your identity to be public for this peer review?** For information about this choice, including consent withdrawal, please see our Privacy Policy .

Reviewer #1: No

Reviewer #2: No

Reviewer #3: No

**Figure resubmission:****Reproducibility:** To enhance the reproducibility of your results, we recommend that authors of applicable studies deposit laboratory protocols in protocols.io, where a protocol can be assigned its own identifier (DOI) such that it can be cited independently in the future. Additionally, PLOS ONE offers an option to publish peer-reviewed clinical study protocols. Read more information on sharing protocols at https://plos.org/protocols?utm_medium=editorial-email&utm_source=authorletters&utm_campaign=protocols

---

## [Editor Report · Decision Letter 1]

Evaluating the diagnostic utility of the slide agglutination method for brucellosis in Saudi Arabia: a retrospective study at International Medical Center, Jeddah, Saudi Arabia

Dear Dr. kaki,

Thank you for submitting your manuscript to PLOS Neglected Tropical Diseases. After careful consideration, we feel that it has merit but does not fully meet PLOS Neglected Tropical Diseases's publication criteria as it currently stands. Therefore, we invite you to submit a revised version of the manuscript that addresses the points raised during the review process.

Please submit your revised manuscript within 60 days Jul 02 2025 11:59PM. If you will need more time than this to complete your revisions, please reply to this message or contact the journal office at plosntds@plos.org. Please include the following items when submitting your revised manuscript:

We look forward to receiving your revised manuscript.

Kind regards,

Richard A. Bowen, DVM PhD

Academic Editor

Elsio Wunder Jr

Section Editor

Shaden Kamhawi

co-Editor-in-Chief

Paul Brindley

co-Editor-in-Chief

**Additional Editor Comments (if provided):**

You do not seem to have submitted a "Response to Reviewers" section, listing the changes you made or why you did not make those changes in response to each of the comments provided by reviewers. You did respond to editorial staff queries but we need the responses to reviewers in order to make a final decision on your manuscript. Thank you in advance.

**Journal Requirements:**

**Reviewers' Comments:**

**Figure resubmission:**
---

## [Editor Report · Decision Letter 2]

Dear Dr kaki,

We are pleased to inform you that your manuscript 'Evaluating the diagnostic utility of the slide agglutination method for brucellosis in Saudi Arabia: a retrospective study at International Medical Center, Jeddah, Saudi Arabia' has been provisionally accepted for publication in PLOS Neglected Tropical Diseases.

Best regards,

Richard A. Bowen, DVM PhD

Academic Editor

Elsio Wunder Jr

Section Editor

Shaden Kamhawi

co-Editor-in-Chief

Paul Brindley

co-Editor-in-Chief

Thank you for your thoughtful consideration of review comments and editing your manuscript for clarity.
